# Social care data and its fitness for integrated health and social care service governance: an exploratory qualitative analysis in the Dutch context

Véronique LLC Bos [ID] ,[1,2] Niek S Klazinga,[1,2] Dionne S Kringos[1,2]

[1]Department of Public and Occupational Health, Amsterdam UMC Location AMC, Amsterdam, The Netherlands
[2]Quality of Care, Amsterdam Public Health Research Institute, Amsterdam, The Netherlands

**Correspondence to**
Véronique LLC Bos;
v.l.bos@amsterdamumc.nl

## ABSTRACT

**Introduction** To date, little is known on how social care data could be used to inform performance-based governance to accelerate progress towards integrated health and social care.

**Objectives and design** This study aims to perform a descriptive evaluation of available social care data in the Netherlands and its fitness for integrated health and social care service governance. An exploratory mixed-method qualitative study was undertaken based on desktop research (41 included indicators), semi-structured expert interviews (13 interviews including 18 experts) and a reflection session (10 experts).

**Setting** The Netherlands; social care is care provided in accordance with the Social Support Act, the Participation Law and the Law for Municipal debt-counselling.

**Results** This study found six current uses for social care data: (a) communication and accountability, (b) monitoring social care policy, (c) early warning systems, (d) controls and fraud detection, (e) outreaching efforts and (f) prioritisation. Further optimisation should be sought through: standardisation, management of data exchange across domains, awareness of the link between registration and financing, strengthening the overall trust in data sharing. The study found five ways the enhanced social care data could be used to improve the governance of integrated health and social care services: (a) cross-domain learning and cooperation (eg, through benchmarks), (b) preventative measures and early warning systems, (c) give insight regarding the quality and effectivity of social care in a broader perspective, (d) clearer accountability of social care towards contracting parties and policy, (e) enable cross-sector data-driven governance model.

**Conclusion** Although there are several innovative initiatives for the optimisation of the use of social care data in the Netherlands, the current social care data landscape and management is not yet fit to support the new policy initiatives to strengthen integrated health and social care service governance. Directions for addressing the shortcomings are provided.

## INTRODUCTION (BACKGROUND AND PROBLEM STATEMENT)

Sound and reliable health and social care information at the right place and at the right

## STRENGTHS AND LIMITATIONS OF THIS STUDY

⇒ In our desk-top research we used an explorative approach and publically available data sources. This can have biased our findings towards finding more data sources aligned with government roles.
⇒ We limited our scope to data sources with a national representation which can have biased results towards finding more aggregate data.
⇒ We minimised these biases by including experts from different levels (micro, meso and macro) and experts from a diverse social care data processing and use background in the interviews and reflection session to enrich the desktop study and validate results.
⇒ Public sources in social care data like the dashboards and monitors used in this study are constantly developing, thus the results of the study are subject to change over time.

time is essential for effective decision making in health systems.[1–3] Integrated health services, described by the WHO as 'health services that are managed and delivered in a way that ensures people receive a continuum of health promotion, disease prevention, diagnosis, treatment, disease management, rehabilitation and palliative care services, at the different levels and sites of care within the health system, and according to their needs throughout their life course' (WHO, 2015), has become an important focus of many health system reforms. As a consequence, performance information presented in the traditional siloes of public health data, healthcare data and social care data form a hindrance for effective governance across the domains of health promotion, disease prevention, diagnosis, treatment, disease management, rehabilitation and palliative care services.[4 5] Challenges in financing (rising (complexity in) health and social care needs and costs), personnel (lack of care professionals) and society (eg, pressure

on solidarity) are jeopardising the sustainability of the healthcare system.[6]

The COVID-19 pandemic has highlighted that in order to effectively monitor and govern our health systems, data from a broad range of sources using demographics, social care data, healthcare data and public health data needs to be systematically integrated. This was demonstrated by the need to monitor information in coherence and including social, behavioural and economic perspectives next to epidemiological considerations.[7] Countries worldwide are struggling with integrating these diverse types of data to establish comprehensive performance intelligence for decision making. Despite data rich health systems, these systems are information poor, indicating an unfulfilled potential of data and digital technologies.[8] Additionally, the COVID-19 pandemic has increased the pressure on the often already overstretched healthcare services and national budgets globally.[9] In order to reduce this pressure policy makers are shifting their attention to health promotion and disease prevention. Strengthening social care is part of this agenda. Investments in social care (eg, housing, income support, nutrition support, care coordination and community outreach) can have a positive impact on the health of citizens and healthcare service needs and use.[10]

Countries and regions worldwide are investing in an integrated health and social care data infrastructure. The main purposes of the data infrastructures differ from primary uses of data (eg, supporting clinical practice) to facilitate a vast range of secondary data uses (eg, research, policy, governance) on regional or national levels. For instance, Finland's Sotkanet, a national health and social care data infrastructure, has more comprehensive data available to inform their decision making from a broader welfare perspective.[11] The affiliated Oulu Self Care Service shows that an integrated data system can also function as an enabler and improve chronic care health outcomes and efficiency through shared use of health and social care data among care providers.[12] New Zealand's integrated health infrastructure (IDI) illustrates that crosscutting data can give insights needed to govern effectively throughout the health and welfare system.[13] [14] This national integrated health infrastructure was for example used for descriptive research to characterise adolescents who experience poor health outcomes.[15] It was also used to define cohorts in the existing population; one example was a cohort of people with chronic diseases to define effects on work and income.[16] Thus, linking social care data to healthcare data shows potential for integrated governance, but there are also known pitfalls. A Canadian study linking administrative social assistance data to healthcare data warns for potential biases due to linkage errors.[17] A study in Dundee, Scotland found that linkage between health and social care data faced challenges around data linkage (eg, use of a shared identifier across organisations), data analysis (eg, understanding missing data) and the need for tacit knowledge (eg, people understanding the data, its limitations and what it represents)

limiting the full exploitation.[18] The Integrated Children's System in the UK initially followed a top-down managerial approach emphasising on accountability of institutional risk instead of user-centredness which had a detrimental effect on professional autonomy.[19] These examples show that a thorough understanding of social care data, its characteristics and use are needed to exploit its potential to gain actionable insights.

The motto of the Dutch Ministry of Health, Welfare and Sport is 'Everyone healthy, fit and resilient'. In the Netherlands multi-stakeholder policy agreements on strategic directions are considered a classical tool for governing the health and welfare sectors. The latest policy changes aim to transform an illness insurance system into a health insurance system. In 2022, Dutch stakeholders in the health and welfare sectors launched multiple integrated policy agreements and programmes to facilitate a more integrate delivery of health and social care services: the Integrated Care Agreement (IZA), the Healthy and Active living Agreement (GALA), the Housing, Support and Care for the Elderly Programme (WOZO) and the Future-proof Care and Welfare Labour Market Programme.[20–23] In these initiatives, cooperation and integration of health and social care services and prevention efforts are highlighted as important ways forward to subdue the pressure on the healthcare system and its staff. The Netherlands has many (N: >450) care partnerships.[24] These are often regional initiatives and alliances that are strengthening their provision of integrated health and care services for better outcomes.[25] The bandwidth of data-driven action in these collaborations is broad. Some collaborations see it as a need to have, others as a nice to have.[24] In 2015, the decentralisation of social care responsibilities increased the municipal responsibilities towards vulnerable individuals and populations. Efforts in integrated care are particularly aimed at for vulnerable individuals and populations.[20–22 26 27] With a high political and regional interest in integrating health and social services, the Netherlands is a relevant country for an explorative study on social care data and its fitness for integrated health and social care service governance.

## General aims and objectives

This study aims to perform a descriptive evaluation of the landscape of social care data in the Netherlands, its current use and how it can be improved. It attempts to explore the fitness for purpose and use of social care data to contribute to integrated health and social care service governance.

## Research questions

1. What are the current uses of social care data in the Netherlands, and how can the usability of this data be enhanced?
2. In what ways can enhanced social care data contribute to the governance of integrated health and social care services?

## RESEARCH METHODS

### Research design and scope

A constructivism exploratory qualitative study design[28] was employed using: (a) desktop research, (b) semi-structured expert interviews and (c) a reflection session. The study conforms to the Consolidated Criteria for Reporting Qualitative Studies, the 32-item checklist for this study can be found in online supplemental material 1. Our research design was grounded in the healthcare performance intelligence pyramid.[1] The hierarchical pyramid defines how measurement through data collection and its translation to indicators is at the base of governance and management through information and knowledge. Action and utilisation is the ultimate end goal and needs its own translation from information and knowledge to practice. Each of these steps needs to consider the regulatory, organisational, political and cultural context.

The scope of this research is social care as provided within the municipal responsibilities in the Netherlands through: (a) the Social Support Act (Wet Maatschappelijke Ondersteuning, WMO); regulating municipal responsibility for supplementing citizens in their care need, (b) the Participation Law (Participatiewet); regulating municipal responsibility for supporting citizens in participating in the society and (c) the Law for Municipal debt-counselling (Wet Gemeentelijke Schuldhulpverlening); regulating municipal responsibility for supporting citizens in financial need. As the focus of this study is on the adult population we have excluded social care as provided by the Youth Act (Jeugdwet); regulating the preventative and mental healthcare for children. As social care was decentralised to municipal level in 2015 in the Netherlands, we have only included documents from beyond this time point in our desktop research.

### Patient and public involvement

Clients, patients and the public have not been involved in the design or conduct of the study. Social care clients were represented in the interviews and the reflection session by the National (social care) Client Council. The results of the study will be disseminated through our network and shared with all study participants.

### Data collection and coding

Our data collection took place from January 2022 to October 2022. The desktop research aimed to get an overview of publicly available social care indicators. We used two starting points: (a) the website vzinfo.nl of the National Institute for Public Health and the Environment commissioned by the Ministry of Health, Welfare and Sport, which provides an overview and metadata of available public health and care information and (b) a social care stakeholder analysis done by Driesten et al. in 2021.[29] Data included in this study was either collected on a national level, or purposefully sampled to represent the national population. Data(sets) that were collected only for a single organisation/municipality or region were excluded from this study. We collected all described indicators from dashboards and reports found via these two starting points that met the scope of this research. An overview of all the collected dashboards and reports included in the desktop research and their references can be found in online supplemental material 2.

The expert interviews aimed to complement the desktop research and highlight current uses of social care data and indicators within the social care domain, to describe ways to optimise social care data (infrastructure), and to highlight potential uses for integrated health and social care service governance. We purposively sampled experts based on the sources found in the desktop research, and gave them the opportunity to refer to other experts in the field of social care data in the Netherlands. The participants were approached via email, phone and LinkedIn. We aimed to have at least two representatives from each level of the healthcare system: micro (social care clients and care professionals), meso (data custodians and data processing organisations) and macro (policy and legal experts) levels. Most interviews were organised digitally in the aftermath of the COVID-19 pandemic and the duration of the interviews was between 33 and 60 min. Three approached participants dropped out without stating a reason, one approached participant could not to participate due to prioritisation issues at that moment. The interview guideline (online supplemental material 3) was grounded in the healthcare performance intelligence pyramid. Before the start of the study, research goals and the research team (department, credentials and occupation) were introduced to the interviewees via email. With the written informed consent of participants, all interviews were audio-recorded and transcribed. Transcripts were emailed to the participants and they were given at least 2 weeks to review and adjust the content of their transcript without limitations. The validated transcripts were used for the thematic analysis.

The reflection session was organised on 12 October 2022 with the following aims: (a) to validate the draft results from the desktop research and semi-structured interviews and (b) to consolidate key findings. It was organised digitally in the aftermath of the COVID-19 pandemic. First, the study team presented the draft results. Second, participants shared their general reflections on the draft results. Finally, the following questions were discussed in-depth: (a) is the outlined landscape of available social care data and indicators, its management and its use recognisable?; (b) how can we optimise the use of available social care data within the social care domain?; (c) how can social care data contribute to public health and healthcare? And vice versa?; and (d) how can social care data be used in governing towards regional integrated care provision? The reflection session was audio recorded with the verbal informed consent of all participants. The draft results from the desktop study and expert interviews were shared with the participants in advance.

## Data analysis

To answer research question 1, we used information retrieved from desktop research and interviews. First, we constructed an overview of all dashboards and documents collected with their data custodian and a qualitative description of their content (online supplemental material 2). Then we abstracted all indicators within the scope of this research and grouped them in their legal silos. Finally, we categorised them into the following subgenres: input, process and output, outcome, and impact and allocated source types (administrative, medical record, registry, survey, direct observation) to each indicator. Transcript pieces that described available social care data in the Netherlands were used to enrich the desktop research by adding additional data sources. We selected and then grouped all pieces of the transcripts from the interviews that described social care data in the Netherlands, its current use and how it can be improved. Then we analysed the transcript pieces using an analytic induction process using key words and statements to define how social care data is currently used and can be improved.

To answer research question 2, pieces of the interview transcripts were grouped that relate to the current and potential use of social care data to improve integrated health and social care service governance. Then an analytic induction process using key words and statements was used to define how social care data was used for integrated health and social care service governance. The draft results of research question 1 and 2 were presented, enriched, clarified and verified in the reflection session.

## RESULTS
### Characteristics of the data sources and informants

The sources included in the desktop research can be found in online supplemental material 2. 13 interviews were conducted with a total of 18 experts including representatives for social care clients and providers, representatives from the municipalities, social care administrative data processors, owners of social care dashboards and indicator reports and experts from the healthcare insurance data field with experience in integrated care projects. Three participants made minor changes to their interview transcript. The changes contained clarifications of the content and additions to the content. No changes to the transcripts were made that affected the results. The interviewee list with background/expertise can be found in online supplemental material 4. The reflection session included ten experts on social care data, including representatives for social care clients (n=1) and providers (n=2), representatives from the municipalities (n=2), social care administrative data processors (n=4, from three different organisations and working with different social care data) and an expert from the healthcare insurance data field with experience in integrated care projects (n=1).

## What social care data is available in the Netherlands and how is it used?
### Measuring social care data: data collection and indicators

The most common data source types used to populate the included social care indicators were administrative data (including the needs assessment done for administrative purposes) and surveys (often samples representative to national population). Indicators on use and costs of services were most common and most often reported on municipal or neighbourhood level. These indicators were often updated once or two times a year. There is a diversity in measurement instruments to measure an individual's needs for social care. However, there is a national guideline for a needs assessment obligatory in order to receive the social care provided. Prediction models (eg, the Social Support Act prediction model) are available. However, one expert mentioned that policymakers find such models difficult to apply due to the lack of data driven personnel and a data driven work culture. Policy makers are also hesitant to use these models as they are by definition incorrect, as they only model reality and can prove to be incorrect in hindsight.

The use of client reported data for decision making is very limited in the social care domain. There is a legal obligation in the Social Support Act for municipalities to carry out a national client experiences measure. Until 2021 this measure consisted of 10 nationally defined questions divided in three themes: (a) access to services, (b) quality of the services provided and (c) the effect on the independence for the client and his/her participation in society. However, since 2021, due to suboptimal use of the measure, municipalities are free to adapt the client experience measure to their needs, either using the 10 defined questions, adding additional questions or using their own survey format. Experts state that this tailoring of the survey by municipalities might be detrimental to cross-municipality comparisons. Besides the national client experience measure, client stories are occasionally used to represent clients' voices to complement the data provided, this is relatively customary in the municipal reporting on social care.

The tables give an overview of all included indicators and their defined indicator and data source type per legal silo.

For the Social Support Act we included 17 indicators (table 1). Most were sourced from survey data or administrative data and gave insight on (potential) inputs or processes and outputs.

For the Participation Law we included 19 indicators (table 2). All were sourced from survey data or administrative data and most gave insight on processes and outputs or outcomes.

For the Law for Municipal debt-counselling we included nine indicators (table 3). All were sourced from survey data or administrative data and most gave insight on processes and outputs.

**Table 1** Social care indicators included in the desktop study within scope of the Social Support Act

| Indicator | Type of indicator | Type of source |
|---|---|---|
| Trend in the number of potential informal carers | (potential) Input | Survey data |
| % vulnerable population | (potential) Input | Administrative/survey/ medical record data |
| % of people with one or more limitation in daily functioning | (potential) Input | Survey data |
| % of people with one or more restrictions in their mobility | (potential) Input | Survey data |
| % of people with restrictions in their eye sight | (potential) Input | Survey data |
| % of people with hearing disabilities | (potential) Input | Survey data |
| Budgeted and actual costs for Social Support Act | Input | Administrative |
| Municipal policy (only customised provisions, mainly customised provisions, as many general as customised provisions, mainly general provisions, only general provisions) of Social Support Act care provision | Input | Survey data |
| Ratio between households with and without care provision | Process and output | Administrative |
| Relative number of households with provisions of care and support by number of provisions | Process and output | Administrative |
| Relative number of households with provisions of care and support by number of provisions per domain | Process and output | Administrative |
| Trend number of inhabitants with Social Support Act care provision | Process and output | Administrative |
| Relative number of inhabitants with Social Support Act care provision by category | Process and output | Administrative |
| Number of inhabitants with Social Support Act care provision | Process and output | Administrative |
| Relative number of inhabitants in sheltered housing by region | Process and output | Survey data/ administrative |
| Number of terminated Social Support Act provisions by reason (total, deceased, planned/anticipated termination, move to other municipality, other/ unknown) | Process and output | Administrative |
| Qualitative reporting of the WMO client experience outcomes | Outcome | Survey data |

WMO, Wet Maatschappelijke Ondersteuning.

### Governance and management in social care data: rules for translation to information and knowledge

National administrative registration standards are in place, for example, the i-WMO standard. However, municipalities have the legal mandate for the social care provision. This mandate also includes how to structure and use (administrative) social care data. This makes national level benchmarking between municipalities difficult, due to a high variability in interpretation and use of the available national standards. The diversity of payment models within and across municipalities also impact the way municipalities structure this data, making this data only interpretable if: (a) within a known context (eg, on municipal level) or (b) aggregated to very general terms where this variation is diminished. Our sources also showed a big variety between municipalities in data processing power, ranging from own data processing departments to support policy decisions to the minimal data processing needed for accountability purposes. This variety in data processing power also exists between social care provider organisations.

Legislation on the use of social care data mainly covers accountability of social provisions (fraud and supervision), the use of the citizen service number for identification, and permission required for use of personal data (privacy). Multiple public and private parties can fulfil the role of data processor in social care in the Netherlands, however often under municipal authority. But the use of social care data for quality improvement of care provision by integrated provider networks, research or policy action is not defined in the legislation. Thus, a weighing between respecting privacy of personal data on the one hand and having data available to deliver adequate social care services or take timely policy action on the other hand, cannot be made.

### Utilisation of social care data: what is done with the acquired knowledge through data

Social care data is mostly used in the *communication and accountability* between care providers and financers (eg, official needs assessments or registration of start and end a care trajectory), and in *monitoring social care policy* on municipal and national levels (eg, monitor on adult health and health of the elderly). There is an *early warning system* on over-indebtedness using non-payments of housing, utilities and healthcare insurance to alert municipalities to a

**Table 2** Social care indicators included in the desktop study within scope of the Participation Law

| Indicator | Type of indicator | Type of source |
|---|---|---|
| Trend in pension age | Input | Survey data |
| Ratio between households with and without care provision | Process and output | Administrative |
| Relative number of households with provisions of care and support by number of provisions | Process and output | Administrative |
| Relative number of households with provisions of care and support by number of provisions per domain | Process and output | Administrative |
| Number of inhabitants with a welfare benefit | Process and output | Administrative |
| Number of inhabitants with a welfare benefit by municipality and reference group | Process and output | Administrative |
| Number of inhabitants with a reintegration facility | Process and output | Administrative |
| Relative number of inhabitants with a reintegration facility by municipality and reference group | Process and output | Administrative |
| Number of unemployed inhabitants in working age | Outcome | Survey data |
| % inhabitants participating in formal work, volunteer work and informal care by sex | Outcome | Survey data |
| % of persons with/without health problems with formal work by age categories | Outcome | Survey data |
| % of the formal work population with/without health problems by age categories | Outcome | Survey data |
| % of net formal work participation by age category and sex | Outcome | Survey data |
| Number of inhabitants of the formal workers working part-time by sex | Outcome | Survey data |
| % Unemployment by age category and sex | Outcome | Survey data |
| Trend % net formal work participation by two age categories and sex | Outcome | Survey data |
| Trend % unemployment by age category and sex | Outcome | Survey data |
| % Unemployed young professionals | Outcome | Survey data |
| SES-WOA score (financial prosperity, level of education and recent employment history of private households) | Impact | Administrative |

potential demand in social care services to prevent over-indebtedness. According to one of the care organisations interviewed this proactive exchange of information has led to the prompt identification of individuals requiring services, thereby minimising the need for involuntary out-of-house placements in the serviced district. Nevertheless, additional insights from interviewed experts underscored that in numerous regions, the transition from data availability to using that data for identification and subsequent action by municipalities, necessary for achieving impact, still requires improvement. Data alone does not lead to action.

Traditionally, data in the social care domain has been used for *controls and fraud detection*, which has led to a distrust in data exchange in the social care domain from the care provider and client perspective. Examples of information used for controls and fraud detection mentioned by interviewees were: (a) financial information on care providers to guide the municipal purchasing risk management strategy and (b) information on which clients have access to Long Term Care Act resources to check whether the Social Support Act resources given are still relevant. Some recent examples rethink this

approach, like the *outreaching efforts* of the Social Insurance Bank (SVB) to track and locate eligible citizens entitled to receive a pension, yet for reasons unknown, have not (yet) applied for it. We have identified an illustrative *prioritisation* case where a social work organisation employed national demographic data in tandem with their client base to evaluate whether they are effectively directing their efforts toward the neighbourhood exhibiting the highest (predicted) need for debt counselling.

### Improving the social care data use
*Data: standardisation of registration*

A high variation between municipalities in social care data, data infrastructure and processing (power) is the result of policy decentralisation efforts aiming to provide care 'closer to home'. Efforts to standardise product codes have already yielded results (from 100.000 product codes in 2015 to about 3.000–4.000 product codes per law during the time of this research). However, some experts interviewed believe that more standardisation in social care data, its infrastructure and its processing can increase comparability and decrease administrative burden on the long run for care provider organisations

**Table 3** Social care indicators included in the desktop study within scope of the Law for Municipal debt-counselling

| Indicator | Type of indicator | Type of source |
|---|---|---|
| Municipal capacity to provide debt counselling services (enough capacity for demand, enough capacity for demand, but sustainability for the future is under pressure, not enough capacity for demand) | Input | Survey data |
| Factors that determine the manner in which early warning signals for over indebtedness are followed up (type of notification, debt size, type of non-payment, age, composition of the household, time passed since notification, postal code) | Input | Administrative/ survey data |
| Number of applications for municipal debt counselling services | Process and output | Survey data |
| Qualitative reporting by cooperating municipalities on the (change in) target group for municipal debt counselling services | Process and output | Survey data |
| Number of early warning signals (one non-payment from one specific provider on one home address) per 1000 inhabitants per municipality | Process and output | Administrative |
| Number of early warning notifications (multiple signals at one address) per municipality | Process and output | Administrative |
| Number of contacts (defined as contact with a reaction, eg, reply to email, opening the door, answering the phone call) resulting from early warning signals | Process and output | Administrative |
| % of inhabitants accepting the offer for debt counselling services | Process and output | Administrative |
| SES-WOA score (financial prosperity, level of education and recent employment history of private households) | Impact | Administrative |

and municipalities alike. We found heterogeneous roles and responsibilities in the coordination and alignment of data processing in social care. It has become difficult to compare even similar indicators due to source and processing variability. Establishing a well-defined governance structure for social care data, along with a robust infrastructure and processing framework, could enhance the stability and interpretability of the available data for more effective utilisation. Standardisation initiatives can provide a coherent framework for data registration, thereby mitigating interpretation challenges associated with the registration process. Representatives from social care organisations emphasise the importance of considering that standardising data registration does not imply a necessity to diminish regional variability in care delivery, especially when diverse regions exhibit distinct care needs.

### Governance and management of social care data across domains and within clear roles for a societal purpose

Data processing is often operationalised per legal and financial silo. Experts state that in order to have a clear picture of the societal needs and impact of social care, insight in performance information across legal and financial silos is needed (eg, the association between over-indebtedness and healthcare use). However, data use across domains is legally and politically restricted in the Netherlands. Additionally, cross-sector data sharing, including across private and public interests, may have more potential for *preventative measures*, but is politically sensitive. For example, in the context of nationally organised debt counselling initiatives, it is noteworthy that study loans exhibit a higher average credit per person compared with consumer credits. However, it is essential to acknowledge that study loans, in contrast to consumer credits, face less transparent monitoring by credit providers, primarily due to political sensitivity. The implementation of these supplementary prevention options could yield positive effects, provided they operate within the requisite boundaries of a robust ethical and legal framework. This ensures alignment between data use and its intended societal purpose while avoiding any potential discrepancies.

Multiple experts state that legislation concerning data exchange and processing ought to prioritise explicit use and purpose rather than merely restricting (while simultaneously facilitating extensive) data processing and integration within specific financial or legal silos. The measurement of service effectiveness and affordability remains constrained, as does the availability of metrics aligned with newly articulated national policy goals, such as the IZA. Currently, there is limited data on the well-being of citizens, including how to sustain healthy lives and identify potential risks to health.

### Awareness of the association between registration and financing mechanisms

We found that indicators in social care frequently rely on administrative data sources, leading to a notable influence from financing structures and incentives on these indicators. One of the examples given in the interviews was in a Participation Law indicator. When an individual transitions to a different role within the same company, the data records this as a job change or switch, with an 'end of trajectory' and a subsequent 'new beginning of trajectory' being registered.

Leveraging this data to gain insights into stability in societal participation was inherently biased, revealing an overemphasis on short-term employer relations.

According to the experts in our study, financing on a population level remains more theoretical than practical. Nonetheless, a shift in financing models could potentially impact data availability within the social care domain. For example, there are municipalities that have started using clustered/population/performance contracts in contracting social care organisations. This means that some municipalities no longer are able to distinguish (parts of) the care provided into potentially relevant subgroups, thus registering entries under 'other', 'unknown' or estimating percentages.

### Improve the trust relationship between municipalities, care providers and clients

The complexity of the social care data infrastructure and processing compromises the transparency towards clients. This has consequences for the trust clients have in the processing of their data. As one expert puts it: 'Unknown makes unloved'. This, together with the history of social care data uses towards control and fraud purposes has deteriorated the relationship between social care financers (eg, municipalities) and clients. To improve this relation a dialogue is needed between clients and social care financers to discuss the different perspectives and bring them together. Both parties must acquire the necessary skills to engage in this dialogue effectively. Additionally, illustrating the processes involved in data exchange and showcasing the potential benefits, as well as the actual outcomes and impact it brings to society, can enhance this relationship. An alternative approach involves shifting the utilisation of data from control and fraud detection purposes to prioritising citizen rights. Data could then be employed to identify individuals entitled to specific social care provisions but are not currently receiving them.

### Potential uses for social care data to improve the governance in integrated care practice and policies

Experts state that the enhanced social care data can contribute to the governance of integrated health and social care services. Five potential uses were stated: (a) cross-domain data (including social care data) can facilitate cross-domain learning and cooperation (eg, through benchmarks), (b) integrating social care data with population demographics, welfare, public health and healthcare data can facilitate preventative measures and create early warning systems, (c) it can give insight regarding the quality and effectivity of social care in a broader perspective, (d) better social care insights can enable a clearer accountability of social care towards contracting parties and policy and (e) it can facilitate parties in a more data-driven governance model.

## DISCUSSION
### Principle findings

This study aimed to investigate the current applications of social care data in the Netherlands, identify areas for improvement, and assess how enhanced social care data can contribute to the governance of integrated health and social care services. We found that there is a rich array of social care data sources and information applications in the Netherlands such as public dashboards and policy monitors. Six current uses for social care data were identified: (a) communication and accountability, (b) monitoring social care policy, (c) early warning systems, (d) controls and fraud detection, (e) outreaching efforts and (f) prioritisation. However, the present state of data and information in social care restricts its use for integrated care governance. Enhancing the data involves initiatives in standardisation, effective management of data exchange across domains, recognising the connection between registration and financing, and bolstering societal trust in data sharing overall. Five potential ways were identified to use this enhanced data to improve the governance in integrated health and social care services: (a) cross-domain learning and cooperation (eg, through benchmarks), (b) preventative measures and early warning systems, (c) give insight regarding the quality and effectivity of social care in a broader perspective, (d) clearer accountability of social care towards contracting parties and policy, (e) enable cross-sector data-driven governance model.

### Strengths and weaknesses of the study

This study is conducted within the Dutch context but can serve as inspiration for other high-income countries transitioning from a fragmented healthcare regulated market to regional governance of integrated care. In our desk-top research we used an explorative approach and publicly available data sources. This can have biased our findings towards finding more data sources aligned with government roles. We limited our scope to data sources with a national representation which can have biased results towards finding more aggregate data. We minimised these biases by including experts from different levels (micro, meso and macro) and experts from diverse backgrounds in social care data processing and usage in the interviews and reflection session to enrich the desktop study and validate results. Public sources in social care data like the dashboards and monitors used in this study are constantly developing, thus the results of the study are subject to change over time. This research was designed to be a starting point and not to exhaust all options. Different stakeholder settings could generate different results. We have limited the bias by intentionally taking available public indicators as a starting point for stakeholders to reflect on and by including a wide variety of stakeholders in our interviews and the reflection session.

## Possible explanations and implications on micro, meso and macro levels

Our results are characteristic for a fragmented healthcare system (OECD, 2022).[30] In these systems legal and financial silos are a barrier for integrated healthcare, public health and social care provision governance[25] (OECD, 2022). In the Netherlands social care responsibilities have been decentralised from the federal to the municipal level, resulting in variations between municipalities regarding data processing and utilisation. There are recent attempts in the policy agenda to integrate care services, expressed by (among other incentives) the IZA, the GALA and the WOZO.[20–23] To effectively implement and oversee these agreements, actionable information tailored to the right individual at the right time is essential. However, our findings reveal that social care data is fragmented, standardised to a limited extent, and offers suboptimal linkage possibilities with various legal and financial care silos. Despite these challenges, there is a current trend toward increased integration, endorsed by a majority of political parties in the Netherlands. The outcomes of the November 2023 elections will additionally influence political support for further reforms in financing arrangements aligned with integrated care ambitions.

To effectively monitor and address health system goals, including quality, accessibility and affordability, it is imperative to link a diverse range of data sources. Statistics Netherlands has demonstrated the feasibility of linkages, although they are frequently employed only at aggregated levels. The recent policy programmes should be accompanied by a monitoring system to monitor progress in achieving the ambitions informing the key stakeholders to support their respective roles. Our findings show that different levels in the system and the different roles represented require different pieces of information in order to support effective decision making. Our findings also indicate that the existing social care data infrastructure falls short in providing individual clients with a comprehensive overview of their integrated health and social care provisions. Additionally, social care providers face challenges in exchanging data with other health and social care professionals. Financers in social care encounter difficulties linking their administrative data to healthcare domain financers, and the current data infrastructure does not adequately support the monitoring of integrated agreements made between different parties. Recently the Dutch government has been working on an overall strategy and legislation (eg, Electronic Data Sharing in Health Care Act) on the information infrastructure in health and social care.[31 32] But the emphasis of the approach is put on the curative healthcare sector. Establishing a governance structure and guidelines is crucial to facilitate the implementation of actionable linkages across domains and within various legal and financial silos.

## Future research

Our research design has provided first-hand insights into the utilisation of social care data, addressing a notable knowledge gap in the existing published literature. Policy ambitions to integrate health and social care requires adequate data and information. We recommend further research to identify the essential data and information exchanges required to support integrated health and social care policy goals. The five potential uses found for social care data to improve the governance of integrated care practice and policies could be a focus for further research that supports implementing the desired policies.

## CONCLUSION

This study identified six current uses for social care data: (a) communication and accountability, (b) monitoring social care policy, (c) early warning systems, (d) controls and fraud detection, (e) outreaching efforts and (f) prioritisation. Further optimisation is recommended through standardisation, effective management of data exchange across domains, increased awareness of the link between registration and financing, and reinforcing societal trust in data sharing. Additionally, the study revealed five ways in which enhanced social care data could be employed to govern integrated health and social care services: (a) fostering cross-domain learning and cooperation (eg, through benchmarks), (b) implementing preventative measures and early warning systems, (c) providing insight into the quality and effectiveness of social care from a broader perspective, (d) enhancing accountability of social care toward contracting parties and policy and (e) enabling a cross-sector data-driven governance model.

**Acknowledgements** The authors thank all citizen representatives, health and social care professionals and data experts who were willing to reflect on social data use in the Netherlands so openly during the interviews and the reflection session.

**Contributors** This research was drafted by VLLCB (female, BSc, MA; PhD candidate), DSK (female, associate professor and principal investigator and educator) and NK (male, full professor of social medicine, MD). The desktop research and expert interviews were executed by VLLCB (experienced in interviews, reflection sessions and qualitative research) in close collaboration and supervision of DSK and NK. The reflection session was moderated by NK and accompanied by DSK and VLLCB who took notes. All authors have read and approved this manuscript. The guarantor of this research is VLLCB.

**Funding** This work was supported by the European Union Funded Marie Sklodowska-Curie Innovative Training Network 'HealthPros' grant number 765141.

**Competing interests** None declared.

**Patient and public involvement** Patients and/or the public were not involved in the design, or conduct, or reporting, or dissemination plans of this research.

**Patient consent for publication** Not applicable.

**Ethics approval** This study involves human participants and was approved by Medical Ethics Review Committee of the Academic Medical Center (reference number: W22_136#22.179). Participants gave informed consent to participate in the study before taking part.

**Provenance and peer review** Not commissioned; externally peer reviewed.

**Data availability statement** Data are available upon reasonable request. The data that support the findings of this study are available on request from the corresponding author (VLLCB) for checks on scientific integrity or quality control

by licensed parties. For other reasons, due to the personal nature of this data, permission of the participants is required.

**ORCID iD**
Véronique LLC Bos http://orcid.org/0000-0002-7447-9662

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
