## [Reviewer comments · BMJ Open]

ARTICLE DETAILS

TITLE (PROVISIONAL)	Social care data and its fitness for integrated health and social care service governance: an exploratory qualitative analysis in the Dutch context
AUTHORS	Bos, Véronique; Klazinga, Niek; Kringos, D

VERSION 1 – REVIEW

REVIEWER	Van Everdingen, Coline Maastricht University, Psychiatry and Neuropsychology department
REVIEW RETURNED	02-Sep-2023

GENERAL COMMENTS	Many health system reforms are aimed at integrating health services. Integrated health services provide a continuum of care in a life course perspective. This paper reports on a qualitative study in the context of the Dutch healthcare and welfare system. The study aims to describe the current social care data and to explore using social care data for integrating health and social care governance. The descriptive evaluation is meant as a starting point. The paper and appendices are finished neatly. The methods section is clear. Yet, making implicit contextual information explicit can make this exploration more interesting. I'll further explain where and how. Opportunities for improvement: 1. Introduction: 'Integrated health services' are defined, referring to the WHO. Starting from the current use of social care data, this paper aims to inform performance-based governance to enable integrating (public, healthcare, and social) care arrangements to maintain or improve health. Effective governance is hampered by extensive care. The introduction provides an extensive summary of the international literature. The authors portray the Dutch context in a functional perspective (multi-stakeholder policy agreements on strategic directions). To portray the background and the problem/challenges, it useful to expand the information on the municipal responsibilities (Wpg; intentions and assumptions of the devolution in 2015, including the appeal made to informal resources). In 2015, the devolution increased the municipal responsibilities towards vulnerable individuals and populations. The latest policy changes aim to transform an illness insurance system into a health insurance. How impacts this the governance? An additional
--

	elaboration on the health vision can assist to depict the challenges of (governance on) integrating healthcare and social care. Please, also add a rights-based perspective. Consider providing more information on the adult population in need of integrated (healthcare and welfare) care. Who are the adults with needs from the separate frameworks? How large is the population with concurrent needs from all the three laws? The authors introduce the term 'dual-track health system management'. Implicit, the paper shows that the municipal welfare services (Wmo, Participatiewet, Gemeentelijke schuldhulpverlening) are included in the health system. This is a bit confusing; please provide clarity. 2. Methods The Youth Act is left aside, as the study is focused to the adult population. Desk research and expert interviews collected data on the sources and current use of social care data. A reflection session was used to validate and consolidate the results. Overall, the methods section is clear. Still, more information on the background and the organizations of the experts is vital. Adding an appendix with the (healthcare and/or social care) expertise and the background of the experts and/or the names of the participating organizations provides clarity on the quality of the collected data. Are all professional workers? What about the participation of the civil society? 3. Tables 1-3: The results section provides three separate tables with the results from the desk research. Please, examine if it's possible to describe/compare the collected social care data (on the Wmo, Participatiewet, Wet Schuldhulpverlening) in a process approach. That would make the information more accessible for an (inter)national audience. That would allow to discuss current and future data use at a higher abstraction level. Then you can transfer tables 1-3 to the Supplemental Materials. 4. Discussion: This study was conducted to inform performance-based governance and foster processes towards integrated care. How relate the results of this first exploration to the performance-based finance strategies in the Netherlands? How impact the additions/changes to the introduction section to your exploration on future use? 5. Limitations The study uses a pragmatic approach. The descriptive evaluation breathes that it is meant as a starting point. The chosen approach to the (health vision underlying the) desk research and the approach of the experts impacts the results. (Would approach with 33% healthcare workers/organizations, 33% social care workers/organizations and 33% civil society citizens have produced different results?)
--	--

	6. Language use The paper contains (very) long sentences. Shortening of the sentences can improve the readability of the text.
--	--

REVIEWER	van der Steen, Jenny Leiden University Medical Center, Public Health and Primary Care
-----------------	--

REVIEW RETURNED	14-Sep-2023
-------------

GENERAL COMMENTS	This is an interesting mixed-methods study on the availability and usefulness of social care data in the Netherlands and its potential when linked to healthcare data. The manuscript is mostly well written and reported using COREQ reporting guidelines (although I could not see the checklist included). Except perhaps for some unclear phrases in the Results section (e.g. page 10 lines 30, 55), use of othering language (“elderly”) and some run-on sentences (e.g. page 13, line 51-56). I have a few suggestions to improve the manuscript. Major 1. In the Results section, the perspective is not always clear; was it from study participants or from the authors? Please clarify or move the authors’ own perspectives to the Discussion. For example, page 10, line 50 (“might be detrimental”), page 14, line 49 (“It should be taken into account”), page 15 line 19 (“legislation... should emphasize”) 2. In the Results and in the Discussion section, six or five potential uses (which number 1-5 in the Results) and six current uses in the Discussion section are repeated in the Conclusions, and are not well integrated. Accountability and warning systems, for example, are part of both lists. How do these compare and what use would be really innovative? Minor, may be helpful to reflect on these points: 3. Detrimental effects of mandating data registration in the UK are mentioned on page 6, line 32. Did the authors find any such concerns in the interviews or discussions in their study? 4. Would you say the Netherlands is a more relevant country to explore social care data integration than the UK, where social and health care delivery in long-term care settings may be more segregated than in the Netherlands (page 7, line 1) 5. Participants could review transcripts and adjust contents (page 8, line 50); did they and what was being revised and was this to increase social desirability or did this lead to losing initial or controversial thoughts? 6. Why were policymakers hesitant to use prediction models (page 10, line 34)? 7. How about use of data for research, in addition to use for quality improvement or policy (page 13, line 35)? 8. WEGIZ? Write in full and add reference (page 18, line 10)
--

VERSION 1 – AUTHOR RESPONSE

Reviewer 1 comments:

Comment	Response
Introduction: 'Integrated health services' are defined, referring to the WHO. Starting from the current use of social care data, this paper aims to inform performance-based governance to enable integrating (public, healthcare, and social) care arrangements to maintain or improve health. Effective governance is hampered by extensive care.	Thank you for the concise summary, no text changes needed.
Introduction: The introduction provides an extensive summary of the international literature. The authors portray the Dutch context in a functional perspective (multi-stakeholder policy agreements on strategic directions). To portray the background and the problem/challenges, it useful to expand the information on the municipal responsibilities (Wpg; intentions and assumptions of the devolution in 2015, including the appeal made to informal resources). In 2015, the devolution increased the municipal responsibilities towards vulnerable individuals and populations. The latest policy changes aim to transform an illness insurance system into a health insurance. How impacts this the governance? An additional elaboration on the health vision can assist to depict the challenges of (governance on) integrating healthcare and social care. Please, also add a rights-based perspective. Consider providing more information on the adult population in need of integrated (healthcare and welfare) care. Who are the adults with needs from the separate frameworks? How large is the population with concurrent needs from all the three laws?	We have added the following sentences to the introduction: "In 2015, the decentralisation of social care responsibilities increased the municipal responsibilities towards vulnerable individuals and populations. Efforts in integrated care are particularly beneficial for vulnerable individuals and populations." and "The latest policy changes aim to transform an illness insurance system into a health insurance system."
Introduction: The authors introduce the term 'dual-track health system management'. Implicit, the paper shows that the municipal welfare services (Wmo, Participatiewet, Gemeentelijke schuldhulpverlening) are included in the health system. This is a bit confusing; please provide clarity.	We have refrained from using the terminology 'dual-track health system management' and have replaced this sentence: "This is demonstrated by the need for a dual-track health system management (monitoring specific and generic information in coherence) and including social, behavioural and economic perspectives next to epidemiologic

	considerations.” With “This was demonstrated by the need to monitor information in coherence and including social, behavioural and economic perspectives next to epidemiologic considerations.”
Methods: The Youth Act is left aside, as the study is focused to the adult population. Desk research and expert interviews collected data on the sources and current use of social care data. A reflection session was used to validate and consolidate the results.	Thank you for the concise summary, no text changes needed.
Methods: Overall, the methods section is clear. Still, more information on the background and the organizations of the experts is vital. Adding an appendix with the (healthcare and/or social care) expertise and the background of the experts and/or the names of the participating organizations provides clarity on the quality of the collected data. Are all professional workers? What about the participation of the civil society?	We have added supplemental material 4: Interviewee list with background/expertise to the manuscript.
Results: The results section provides three separate tables with the results from the desk research. Please, examine if it’s possible to describe/compare the collected social care data (on the Wmo, Participatiewet, Wet Schuldhulpverlening) in a process approach. That would make the information more accessible for an (inter)national audience. That would allow to discuss current and future data use at a higher abstraction level. Then you can transfer tables 1-3 to the Supplemental Materials.	Although we share the concern of the reviewer, our preference would be to keep the tables as part of the main text, because they are a good illustration of the heterogeneity of measures per legislative silo. They also provide overall transparency and the detail necessary to come to meaningful results. However we have readjusted the result section with a descriptive paragraph to every table to enhance the readability for an international audience.
Discussion: This study was conducted to inform performance-based governance and foster processes towards integrated care. How relate the results of this first exploration to the performance-based finance strategies in the Netherlands? How impact the additions/changes to the introduction section to your exploration on future use?	We have added the following sentence to the discussion section: “Despite these challenges, there is a current trend toward increased integration, endorsed by a majority of political parties in the Netherlands. The outcomes of the November 2023 elections will additionally influence political support for further reforms in financing arrangements aligned with integrated care ambitions.”
Limitations: The study uses a pragmatic approach. The descriptive evaluation breathes that it is meant as a starting point. The chosen approach to the (health vision underlying the) desk research and the approach of the experts impacts the results. (Would approach with 33% healthcare workers/organizations, 33% social care workers/organizations and 33% civil	We have added the following sentence to the limitations section: “This research was designed to be a starting point and not to exhaust all options. Different stakeholder settings could generate different results. We have limited the bias by intentionally taking available public indicators as a starting point for stakeholders to reflect on and by including a wide variety of

society citizens have produced different results?)	stakeholders in our interviews and the reflection session.”
General: The paper contains (very) long sentences. Shortening of the sentences can improve the readability of the text.	We have shortened lengthy sentences to improve readability.

Reviewer 2 comments:

Comment	Response
General: This is an interesting mixed-methods study on the availability and usefulness of social care data in the Netherlands and its potential when linked to healthcare data. The manuscript is mostly well written and reported using COREQ reporting guidelines (although I could not see the checklist included). Except perhaps for some unclear phrases in the Results section (e.g. page 10 lines 30, 55), use of othering language (“elderly”) and some run-on sentences (e.g. page 13, line 51-56). I have a few suggestions to improve the manuscript.	Thank you for the compliments. We have improved unclear phrases and shortened lengthy sentences to improve readability.
Results (major): In the Results section, the perspective is not always clear; was it from study participants or from the authors? Please clarify or move the authors’ own perspectives to the Discussion. For example, page 10, line 50 (“might be detrimental”), page 14, line 49 (“It should be taken into account”), page 15 line 19 (“legislation... should emphasize”)	Authors own perspectives are not stated in the result section. We have clarified the examples given by the reviewer in the manuscript: “Data experts state that this tailoring of the survey by municipalities might be detrimental to cross-municipality comparisons.”, “Representatives from social care organizations emphasize the importance of considering that standardizing data registration does not imply a necessity to diminish regional variability in care delivery, especially when diverse regions exhibit distinct care needs.”, “Multiple experts state that Legislation concerning data exchange and processing ought to prioritize explicit use and purpose rather than merely restricting (while simultaneously facilitating extensive) data processing and integration within specific financial or legal silos.”
Results/Discussion/Conclusion (major): In the Results and in the Discussion section, six or five potential uses (which number 1-5 in the Results) and six current uses in the Discussion section are repeated in the Conclusions, and are not well integrated. Accountability and warning systems, for example, are part of both lists. How do these compare and what use would be really	We have rephrased our research questions and wording throughout the manuscript to clarify the three different sections of our study and their relatedness as follows: 1. What are the current uses of social care data in the Netherlands, and how can the usability of this data be enhanced? 2. In what ways can enhanced social care data

innovative?	contribute to the governance of integrated health and social care services?
Results/Discussion (minor): Detrimental effects of mandating data registration in the UK are mentioned on page 6, line 32. Did the authors find any such concerns in the interviews or discussions in their study?	We did not find a detrimental effect on professional autonomy as social care data in the Netherlands is still first and foremost owned by the professionals themselves. However we did find a detrimental effect on trust in clients when social care data was used foremost as a fraud detection tool. As stated in the results: “Traditionally, data in the social care domain has been used for controls and fraud detection, which has led to a distrust in data exchange in the social care domain from the care provider and client perspective.”
Introduction (minor): Would you say the Netherlands is a more relevant country to explore social care data integration than the UK, where social and health care delivery in long-term care settings may be more segregated than in the Netherlands (page 7, line 1)	Both countries are facing sustainability issues related to financing and workforce shortages that drive the need for integration of health care and social care. In the introduction we explain why the Netherlands is a relevant country to study this research question. “With a high political and regional interest in integrating health and social services, the Netherlands is a relevant country for an explorative study on social care data and its fitness for integrated health and social care service governance.”
Methods (minor): Participants could review transcripts and adjust contents (page 8, line 50); did they and what was being revised and was this to increase social desirability or did this lead to losing initial or controversial thoughts?	We have added the following sentences to the manuscript: “Three participants made minor changes to their interview transcript. The changes contained clarifications of the content and additions to the content. No changes to the transcripts were made that affected the results.”
Results (minor): Why were policymakers hesitant to use prediction models (page 10, line 34)?	We have replaced the following sentence to the manuscript: “Prediction models (e.g. the Social Support Act prediction model) are available, but in the interview with an expert on the prediction models, it was mentioned that policymakers are still hesitant to use prediction modelling to inform their policy actions.” With “Prediction models (e.g. the Social Support Act prediction model) are available. However, one expert mentioned that policymakers find such models difficult to apply due to the lack of data driven personnel and a data driven work culture. Policy makers are also hesitant to use these models as they are by definition incorrect, as they only model reality and can prove to be incorrect in hindsight.”

Results (minor): How about use of data for research, in addition to use for quality improvement or policy (page 13, line 35)?	We have adjusted the following sentence in the manuscript: “But the use of social care data for quality improvement of care provision by integrated provider networks or policy action is not defined in the legislation.” To “But the use of social care data for quality improvement of care provision by integrated provider networks, research or policy action is not defined in the legislation.”
Discussion (minor): WEGIZ? Write in full and add reference (page 18, line 10)	We have written in full e.g.: Electronic Data Sharing in Health Care Act and have added the reference.

VERSION 2 – REVIEW

REVIEWER	Van Everdingen, Coline Maastricht University, Psychiatry and Neuropsychology departmen
REVIEW RETURNED	01-Jan-2024

GENERAL COMMENTS	The authors produced substantive improvements to the manuscript, for instance in the formulation of the research questions. They provide a more concrete, recognizable description of the results. Finally, the language use and readability are improved. Still, minor revisions are essential for finalizing the manuscript. First impression of the revised manuscript: Many health system reforms are aimed at integrating health services. Integrated health services provide a continuum of care in a life course perspective. This paper reports on a qualitative study in the context of the Dutch healthcare and welfare system. The study aims to describe the current social care data and to explore using social care data for integrating health and social care governance. The authors produced substantive improvements to the manuscript, for instance in the formulation of the research questions. They provide a more concrete, recognizable description of the results. Finally, the language use and readability are improved. Still, minor revisions are essential for finalizing the manuscript. Opportunities for improvement:  Introduction & discussion section: The descriptive evaluation is meant as a starting point. The paper provides an appraisal of the fitness of social care data for integrated health and social care service governance. Aiming at integrating health and social care service governance is attended with challenges due to differing visions of ‘health’ and ‘care’. Therefore, I encouraged the authors to make implicit contextual information explicit.
---

	The interviewee list presented in Supplemental Material 4 presents and overview of the participating organizations and platforms. The expertise of the participating experts in social care is well-documented; their health expertise is less documented. It also remains unclear how the microlevel is represented. Still, in the introduction an additional elaboration on the health vision can assist to depict the challenges of (governance on) integrating healthcare and social care. This enables to better the discussion of the value and limitations of this study in the discussion section. Regarding the discussion: The paper is focused to gain actionable insights. What else can the authors say, based on the presented data at this starting point, on the character of essential processes and the direction of the way to go? 2. Discussion section & limitations For pragmatic reasons, the authors focused the scope of the descriptive evaluation to data sources with a national representation. Still, the recent devolution increased the municipal responsibilities, while the right care at the right time is largely organized in daily life contexts. Extension of the discussion and/or limitations at this point is recommended. 3. Textual improved My final remark relates to a sentence in the revised version: "Efforts in integrated care are particularly beneficial for vulnerable individuals and populations." Recommendation to replace 'are particular beneficial for' with 'are particularly aimed at'.
--	---

VERSION 2 – AUTHOR RESPONSE

Reviewer 1 comments:

Comment	Response
General: Many health system reforms are aimed at integrating health services. Integrated health services provide a continuum of care in a life course perspective. This paper reports on a qualitative study in the context of the Dutch healthcare and welfare system. The study aims to describe the current social care data and to explore using social care data for integrating health and social care governance. The authors produced substantive improvements to the manuscript, for instance in the formulation of the research questions. They provide a more concrete, recognizable description of the results. Finally, the language use and readability are improved. The descriptive evaluation is meant as a starting point. The paper provides an appraisal of the fitness of social care data for integrated health and social care service governance. Aiming at integrating health and social care service governance is attended with challenges due to differing visions of 'health' and 'care'. Therefore,	Thank you for the concise summary, no text changes needed.

I encouraged the authors to make implicit contextual information explicit.	
The interviewee list presented in Supplemental Material 4 presents and overview of the participating organizations and platforms. The expertise of the participating experts in social care is well documented; their health expertise is less documented. It also remains unclear how the microlevel is represented.	The literature so far has mainly focused on the health care perspective. In this study we have put the emphasis on the social care domain and have purposively selected experts on this domain. In this study we used representatives of clients as a proxy for the micro level, as stated in our PPI Statement: "Social care clients were represented in the interviews and the reflection session by the National (social care) Client Council." and mentioned in Supplemental Material 4.
Still, in the introduction an additional elaboration on the health vision can assist to depict the challenges of (governance on) integrating healthcare and social care. This enables to better the discussion of the value and limitations of this study in the discussion section.	We have added an urgency statement and the motto of the Ministry of Health, Welfare and Sport of the Netherlands to the introduction: "Challenges in financing (rising (complexity in) health and social care needs and costs), personnel (lack of care professionals) and society (e.g. pressure on solidarity) are jeopardizing the sustainability of the health care system." And "The motto of the Dutch Ministry of Health, Welfare and Sport is "Everyone healthy, fit and resilient.""
Regarding the discussion: The paper is focused to gain actionable insights. What else can the authors say, based on the presented data at this starting point, on the character of essential processes and the direction of the way to go?	We have emphasised the direction of the way to go in our discussion by adding the following sentence in the discussion section: "Our findings show that different levels in the system and the different roles represented require different pieces of information in order to support effective decision making." This addition emphasizes the actionability aspect for performance information for its user and intended usage (meeting a specific information need).
Discussion & limitation section: For pragmatic reasons, the authors focused the scope of the descriptive evaluation to data sources with a national representation. Still, the recent devolution increased the municipal responsibilities, while the right care at the right time is largely organized in daily life contexts. Extension of the discussion and/or limitations at this point is recommended.	We have stated in the limitation section how we tackled this limitation: "In our desk-top research we used an explorative approach and publicly available data sources. This can have biased our findings towards finding more data sources aligned with government roles. We limited our scope to data sources with a national representation which can have biased results towards finding more aggregate data. We minimized these biases by including experts from different levels (micro, meso and macro) and experts from diverse backgrounds in social care data processing and usage in the interviews and reflection session to enrich the desktop study and validate results."
Textual improvements: My final remark relates to a sentence in the revised version: "Efforts in integrated care are particularly beneficial for vulnerable individuals and populations." Recommendation to replace 'are particular beneficial for' with 'are particularly aimed at'.	We have replaced 'are particular beneficial for' with 'are particularly aimed at' in the revised manuscript.